# Cobalt-catalyzed cross-electrophile coupling of alkynyl sulfides with unactivated chlorosilanes

Donghui Xing[1], Jinlin Liu[1], Dingxin Cai[1], Bin Huang[1], Huanfeng Jiang [1] & Liangbin Huang [1]✉

Herein, we disclose a highly efficient cobalt-catalyzed cross-electrophile alkynylation of a broad range of unactivated chlorosilanes with alkynyl sulfides as a stable and practical alkynyl electrophiles. Strategically, employing easily synthesized alkynyl sulfides as alkynyl precursors allows access to various alkynylsilanes in good to excellent yields. Notably, this method avoids the utilization of strong bases, noble metal catalysts, high temperature and forcing reaction conditions, thus presenting apparent advantages, such as broad substrate scope (72 examples, up to 97% yield), high Csp-S chemo-selectivity and excellent functional group compatibility (Ar-X, X = Cl, Br, I, OTf, OTs). Moreover, the utilities of this method are also illustrated by downstream transformations and late-stage modification of structurally complex natural products and pharmaceuticals. Mechanistic studies elucidated that the cobalt catalyst initially reacted with alkynyl sulfides, and the activation of chlorosilanes occurred via an $S_N2$ process instead of a radical pathway.

Transition-metal catalyzed cross-electrophile coupling (XEC) has become a prominent method for synthesizing molecules, attracting significant attention in the field[1–6]. These reactions offer several advantages, including the avoidance of pre-generation and handling of sensitive organometallic reagents, high efficiency in terms of steps, and excellent compatibility with various functional groups compared to traditional methods. The groundbreaking studies by Weix and Gosmini using nickel and cobalt catalysis paved the way for advancements in C-C bond formation[7–16]. Despite the significant progress in C-C bond formation through XEC, there remains a significant gap in the development of XEC reactions between carbon electrophiles and readily available chlorosilanes for producing organosilicon compounds, highlighting a pressing need for further research in this area[17–19].

Organosilicon compounds have unique properties that are broadly applicable to synthetic chemistry, medicinal chemistry, materials science and other fields[20–23]. Very recently, the Shu, Oestreich and other group respectively reported several elegant Ni-catalyzed cross-electrophile protocols to form C-Si bond, in which a variety of aryl/vinyl/alkyl electrophiles underwent a highly efficient coupling with activated chlorosilanes (R = vinyl, H, Fig. 1a)[24–29]. Subsequently, Fe- and Cr-catalyzed XEC of activated chlorosilanes had also been developed using different catalytic systems[30,31]. Despite formidable advances, no examples of XEC of alkynyl electrophiles with silyl electrophiles by transition-metal catalysis to form Csp-Si bond, have been reported until now.

We speculated that there were mainly two reasons. Firstly, alkynyl electrophiles, such as bromoalkynes, were too active to undergo homocoupling to form 1,3-diynes instead of the cross-coupling products[32–34]. Meanwhile, the low reactivity and the high bond dissociation energy (BDE) of Si-Cl bond[35] (113 kcal/mol) compared to the C-Cl bond[36] (84 kcal/mol) were adverse to its XEC with reactive alkynyl electrophiles. Secondly, the low nucleophilicity of Csp-M intermediate compared with aryl- or alkyl-M species exhibited low reactivity towards chlorosilanes. To verify this hypothesis, bromoalkyne **1-a** and TMSCl were utilized as the starting materials to conduct preliminary experimental exploration (Fig. 1a)[24–26,28]. Mainly the dimer product **4-a** was obtained due to the mismatched reaction rates. Thus, utilizing an

[1]Key Laboratory of Functional Molecular Engineering of Guangdong Province, School of Chemistry and Chemical Engineering, South China University of Technology, Guangzhou, China. ✉e-mail: huanglb@scut.edu.cn

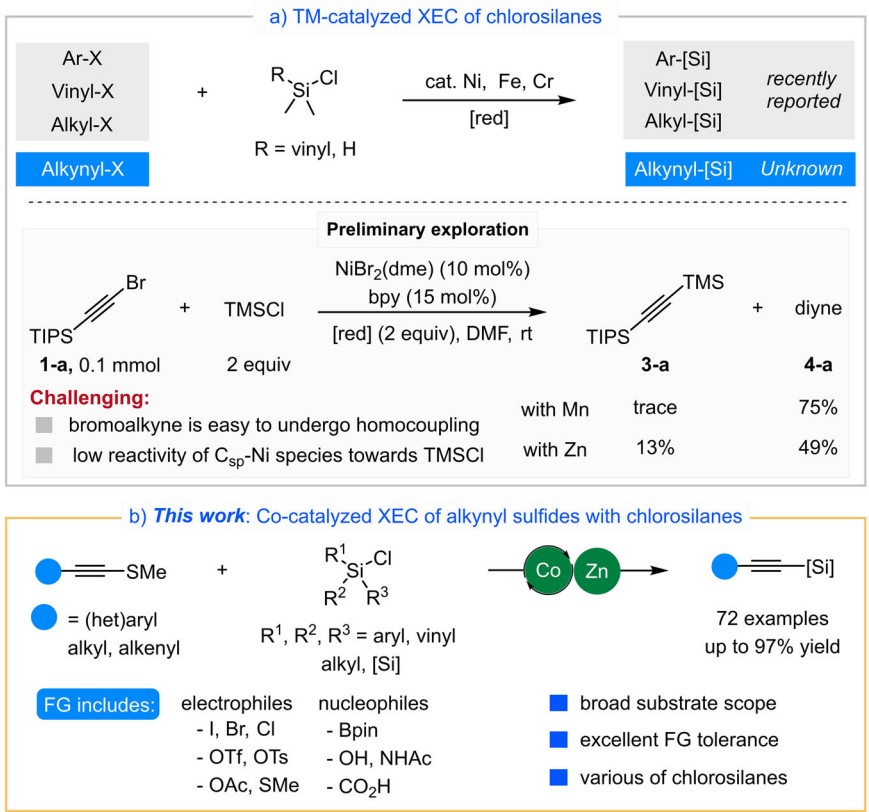

**Fig. 1 | Cross-electrophile C − Si coupling reactions. a** TM-catalyzed XEC of chlorosilanes. **b** This work: XEC of alkynyl sulfides with chlorosilanes.

appropriate alkynyl electrophile and metal catalyst was the key to achieve the XEC between alkynyl electrophiles and chlorosilanes.

Easily synthesized and air-stable alkynyl sulfides[37,38] had sporadically been utilized as alkynyl electrophiles for metal catalyzed cross-coupling[39,40], while was unknown for XEC. In this report, we disclose a highly efficient cobalt-catalyzed cross-electrophile alkynylation of a broad range of unactivated chlorosilanes with alkynyl sulfides as a stable and practical alkynyl electrophiles (Fig.1b). This method generates various alkynylsilanes (72 examples, up to 97% yield) under mild condition, which avoids the use of strong bases, noble metal catalysts and high temperature, providing an alternative and complementary approach[41–43].

## Results

### Investigation of reaction conditions

We started with our investigations by using alkynyl sulfide alkynyl sulfide **1a** and commercially available TMSCl as the model substrates. Extensive examination of the reaction parameters revealed that the combination of 10 mol% of $CoI_2$ as the catalyst and 15 mol% of **L8** (tpy) as ligand with zinc as the reductant in DMF at room temperature was the optimal conditions to deliver the desired product **3a** in 85% isolated yield (Table 1, entry 1, see Supplementary Tables for details). Different Co(II) precursors were tested, $CoCl_2$, $CoBr_2$ and $Co(acac)_2$ gave the product **3a** in high yields but with less efficiency (entries 2-3). Meanwhile, the cross-selectivity decreased dramatically and a certain amount of 1,3-diyne was obtained when $NiI_2$ or $CrCl_2$ was utilized instead of $CoI_2$ as the catalyst (entry 4)[24,28,31]. The alkynyl sulfide **1a** was fully recovered in the presence of iron salt precatalyst (entry 5)[30]. Solvent screening demonstrated that acetonitrile (MeCN) or tetrahydrofuran (THF) shut down the reaction completely (entry 6). The replacement of Zn to Mn resulted no conversion of this reaction (entry 7). When 1 equiv of $ZnI_2$ was added together with Mn, the reaction generated the product **3a** in 67% yield (entry 8). This indicated that Zn

not only acted as reductant, but also the in-situ formed zinc salt as a thiophilic reagent to decrease the inhibition of the leaving sulfide part to catalyst[44]. Control experiments showed that the stoichiometric reductant zinc, the cobalt catalyst and ligand were all essential for this transformation (entry 9). The effect of ligand indicated that bidentate ligands such as bipyridine and phenanthroline derivatives, though were not as good as terpyridine **L8**, still gave the desired product in reduced yields. Conversely, the sterically hindered ligand **L7** failed to give the desired product **3a**.

### Substrate scope

With the optimized reaction conditions in hand, we explored the scope of alkynyl sulfides for this cobalt-catalyzed XEC with TMSCl or $PhMe_2SiCl$, and the results were summarized in Fig. 2. Alkynyl sulfides with different electronic properties or substitution patterns (**3a-t**) were tolerated. Firstly, alkynyl sulfides with diverse electron-donating groups at the *para*-position of the aryl moiety, including ether (**3a-c**), -OAc (**3 f**), -OCF$_3$ (**3 g**), -OTIPS (**3 h**) and -NHAc(**3j**), were converted into the corresponding products in good yields (41–93%). The method also demonstrated high Csp-S chemo-selectivity as reactions with an alkynyl sulfide containing an aryl sulfide moiety to give the product **3i** in 91% yield[45–48]. Additionally, the electron-withdrawing groups, such as -F (**3k-l**), -CN (**3 m**), -CO$_2$Me (**3n**), -CONMe$_2$ (**3o**), and -CO$_2$H (**3p**) were all compatible with the method and delivered the desired products in moderate to excellent yields (55-93%). Although the presence of a terminal alkene or alkyne was challenging to the silylation procedure[49,50], the corresponding products **3q** and **3r** were successfully obtained in 75% and 88% yield, respectively. Moreover, a free alcohol (**3 s**) or amine (**3t**), which could react directly with chlorosilanes, was well tolerated under the standard conditions. When alkyl or alkenyl substituted alkynyl sulfides were selected as substrates, the alkynylsilylation reactions occurred as expected to afford the corresponding products in excellent yields (**3u-y**). The reactions of alkynyl

**Table 1 | Optimization of the XEC of alkynyl sulfide with TMSCl**

| Entry | Variation from standard conditions[a] | Yield of 3a [%][b] | Yield of 1a-1 [%][b] |
|---|---|---|---|
| 1 | none | **87(85)** | n.d. |
| 2 | CoCl$_2$ or CoBr$_2$ as catalyst | 69 or 73 | n.d. |
| 3 | Co(acac)$_2$ as catalyst | 70 | trace |
| 4 | NiI$_2$ or CrCl$_2$ as catalyst | 45 or 29 | 19 or 37 |
| 5 | Fe(acac)$_2$ as catalyst | n.d. | n.d. |
| 6 | MeCN or THF as solvent | trace | 13 or 31 |
| 7 | Mn instead of Zn | n.d. | trace |
| 8 | Mn instead of Zn, ZnI$_2$ (1 equiv) | 67 | n.d. |
| 9 | w/o CoI$_2$ or Zn or **L8** (tpy) | n.d. | n.d. |

**effect of ligand**

[a] Reaction conditions: **1a** (0.1 mmol), TMSCl (0.2 mmol), CoI$_2$ (10 mol%), **L8** (15 mol%), Zn (2.0 equiv), DMF (1.0 mL), rt, 4 h. [b] The yields were determined by GC using dodecane as the internal standard; Isolated yield was shown in parentheses; n.d. = no detected; w/o = without.

sulfides containing polyaromatic or heteroaromatic ring proceeded smoothly to produce alkynylsilanes (**3z-aj**) in moderate to good yields, including fluorene (**3z**), naphthalene (**3aa**) thiophene (**3ab, 3ac**), pyridine (**3ad**), quinoline (**3ae**), benzofuran (**3af**), indole (**3ag**), benzothiophene (**3ah**), carbazole (**3ai**) and dibenzofuran (**3aj**). Given the broad substrate scope and good functionality tolerance of our strategy, we further explored this transformation to for late-stage functionalization with more complex molecules. Alkynyl sulfides derived from Donepezil (**3ak**), Desloratadine (**3al**), Citronellol (**3am**), DL-Menthol (**3an**), Gemfibrozil (**3ao**), and Cholesterol (**3ap**), were all suitable substrates, providing the corresponding products in 41-87% yields.

Next, we explored the reactivity of chlorosilanes by varying the substituents on silicon for these transformations (Fig. 3). A series of chlorodimethyl(alkyl)silanes with varied chain lengths and steric properties reacted with alkynyl sulfide **1a**, giving the corresponding alkynylsilylation products in moderate to high yields. For instance, chlorodimethyl(ethyl) silane (**4a**), chlorodimethyl(butyl) silane (**4b**), chlorodimethyl(3-phenylpropyl) silane (**4c**) and chlorodimethyl-(butanenitrile) silane (**4d**) furnished the desired products in 83% to 96% yields. Chlorosilanes bearing steric substituents, such as diaryl (**4 f**), isopropyl (**4 g**), cyclohexanyl (**4 h**) and TMS (**4i**), all worked well for this cobalt-catalyzed system, affording the corresponding

products with good efficiency. The presence of allylic and vinyl functionalities on the chlorosilanes was well-tolerated, offering opportunities for further diversification of the resulting products (**4j-l**). It is worth noting that several chlorotrialkylsilanes with larger steric hindrance were also tested, yielding the corresponding products in the range of 69% to 71% (**4m-n**).

To further exhibit the excellent chemo-selectivity of alkynyl sulfides as alkynyl electrophiles for these reactions, we conducted the XEC with alkynyl sulfides containing various electrophilic and nucleophilic reaction sites and the results were summarized in Fig. 4. These studies demonstrated that this Co-catalyzed silylation of alkynyl sulfides offered orthogonal selectivity towards other well-established cross-coupling methods. Aryl electrophiles, including chloride (**3aq**), bromides (**3ar–as**), iodide (**3at**), tosylate (**3au**), and triflates (**3av, 3aw**), were all tolerated under the standard conditions. In addition, alkynyl sulfides containing additional electrophilic functional groups were converted to alkynylsilanes in good yields (**3ax-bc**), which provided the chance to further modify these molecules. An alkynyl sulfide containing an alkyl chloride moiety was selectively silylated at the Csp-S site (**3bd**). Alkynylsilane containing a boronic ester group (**3be**) was constructed in 85% yield for this Co-catalyzed XEC.

Alkynylsilanes are synthetically useful intermediates, which have been widely used in synthetic chemistry[51–59]. To show the synthetic

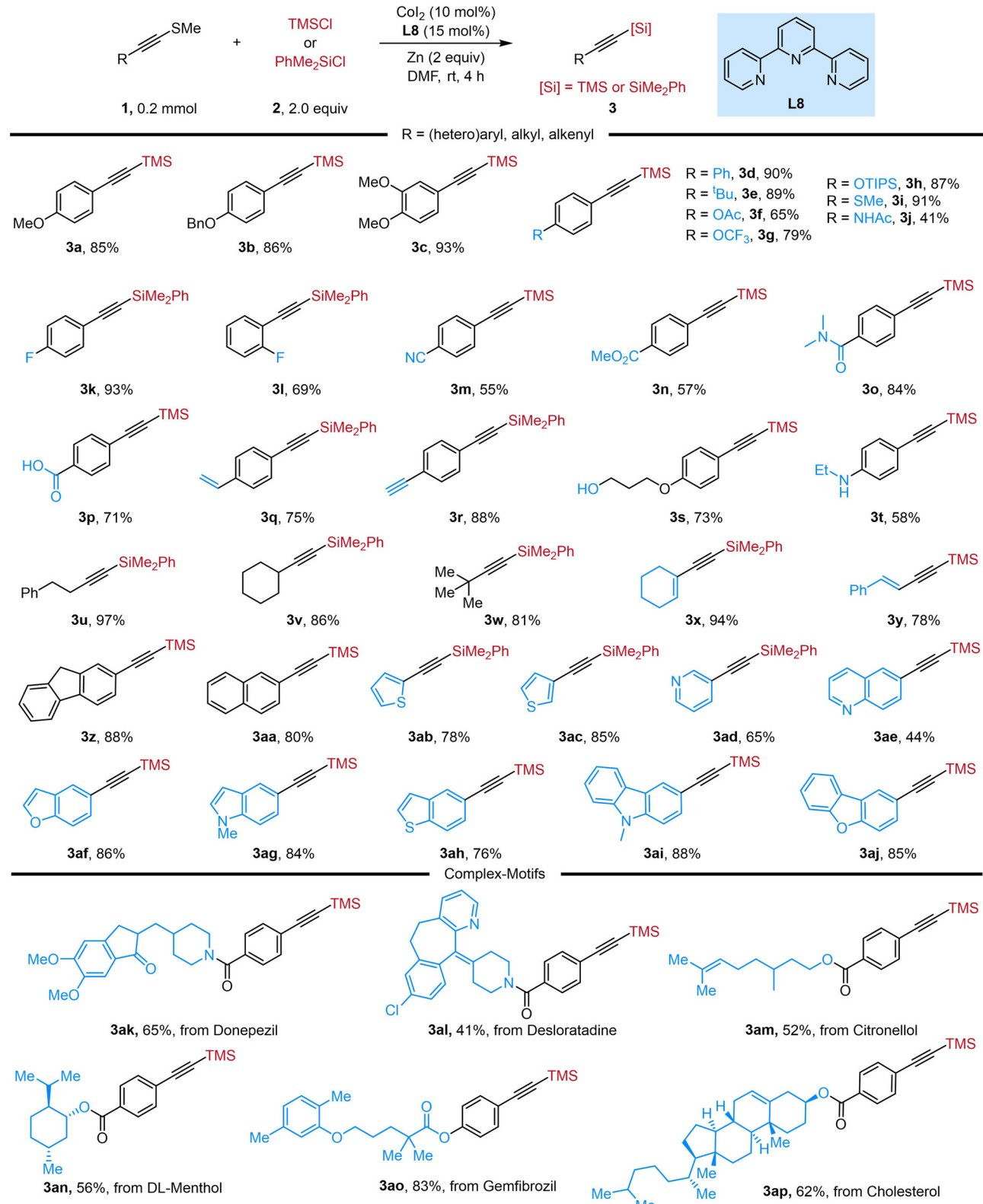

**Fig. 2 | Substrate scope of alkynyl sulfides.** Reaction conditions: **1** (0.2 mmol), **2** (2.0 equiv), CoI₂ (10 mol%), **L8** (15 mol%), Zn (2.0 equiv), DMF, rt, 4 h.

utility of this protocol, we conducted a series of further transformation of the desired alkynylsilane. First, a gram-scale reaction of alkynyl sulfide **1ay** with a reduced catalyst loading (5 mol%) to generate alkynylsilane **3ay** in 71% yield. Tetrabutylammonium fluoride catalyzed the addition of **3ay** to trifluoromethyl ketone, producing the CF₃-substituted tertiary propargylic alcohol **5a** in moderate yield (Fig. 5a-a). A

Cu-catalyzed three-component coupling reaction of **3ay**, *o*-hydroxybenzaldehyde and amine, formed the corresponding benzofuran **5b** in 76% yield, involving an intramolecular *5-exo-dig* cyclization (Fig. 5a, b). A copper-catalyzed three-component coupling reaction of alkynylsilane **3ay**, aldehyde and amine generated the propargylic amine **5c** in 84% yield (Fig. 5a–c). A ZnCl₂-catalyzed Diels-Alder/retro-

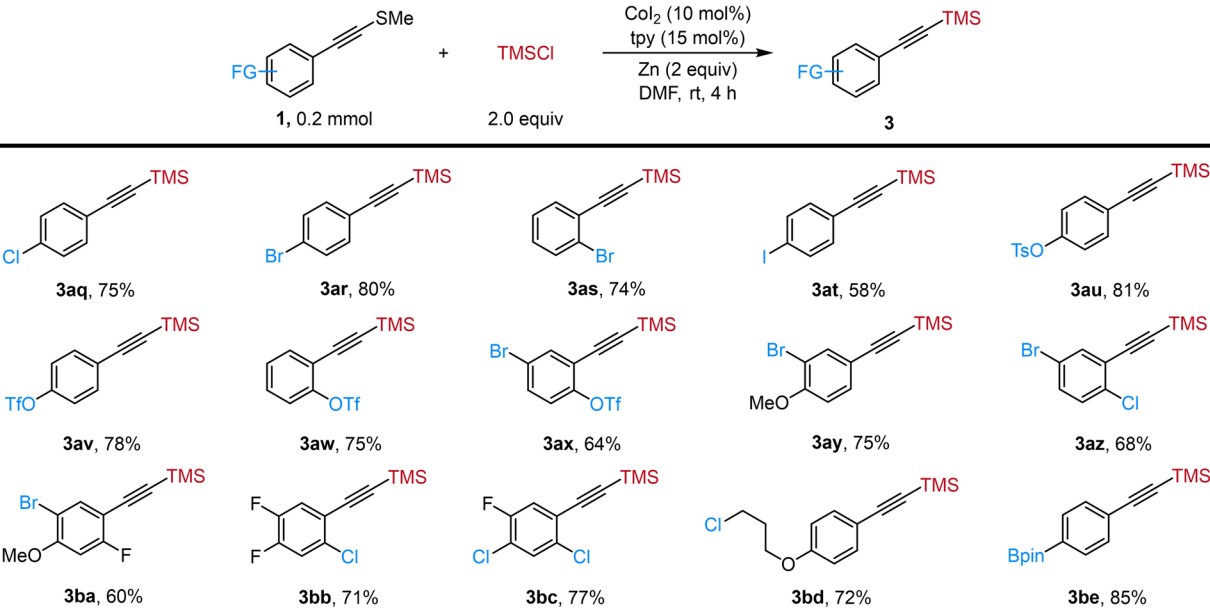

**Fig. 3 | Substrate scope of chlorosilanes.** Reaction conditions: **1a** (0.2 mmol), **2** (2.0 equiv), CoI$_2$ (10 mol%), **L8** (15 mol%), Zn (2.0 equiv), DMF, rt, 4 h.

**Fig. 4 | Chemoselectivity C(sp)-Si couplings.** Reaction conditions: **1** (0.2 mmol), TMSCl (2.0 equiv), CoI$_2$ (10 mol%), tpy (15 mol%), Zn (2.0 equiv), DMF, rt, 4 h.

Diels-Alder reaction between electron-deficient 2-pyrone and alkynylsilane **3ay** enabled the synthesis of arylsilane **5d** in 73% yield (Fig. 5a–d). The gold-catalyzed reaction of *o*-alkynylbenzaldehydes with alkynylsilane **3ay** gave naphthylsilane **5e** in 49% yield (Fig. 5a–e). And the Cobalt(I)–diphosphine catalyzed dehydrogenative annulation between alkynylsilane **3ay** and salicylaldehyde afforded the corresponding 2-aryl-3-silylchromone **5f** in 69% yield (Fig. 5a–f). Moreover, a Co-catalyzed regioselective [3 + 2] annulation of *ortho*-functionalized arylboronic acid with **3ay** gave the corresponding cyclized products **5g** and **5h** in 81% and 65% yield respectively Fig. 5ag, h). To further demonstrate the synthetic utility of this developed methodology, the efficient synthesis of 6-fluoroflavone, an anti-rhinovirus agent, was carried out using current method as the key step (Fig. 5b). Finally, we

extended our method to several examples of the synthesis of alkynylgermanes, all of which had good yields (Fig. 5c).

To elucidate the underlying mechanism of these reactions, a set of experiments was conducted as shown in Fig. 6. Firstly, to reveal whether a radical process was involved in the activation of R$_3$Si-Cl, several radical acceptors (3.0 equiv) were added to the standard reaction conditions and **4e** was obtained in 69-82% yields with no sign of the expected side product **7** from radical trapping (Fig. 6a). A radical clock experiment of chlorosilane **2o** was performed, and the directly silylated product **4o** was obtained in 87% yield with no cyclized derivative detected (Fig. 6b). These results suggest that Si-Cl is not activated through a radical process. Subsequently, alkynyl sulfide **1a** was transformed to the homodimer (diyne) product **1a-1**

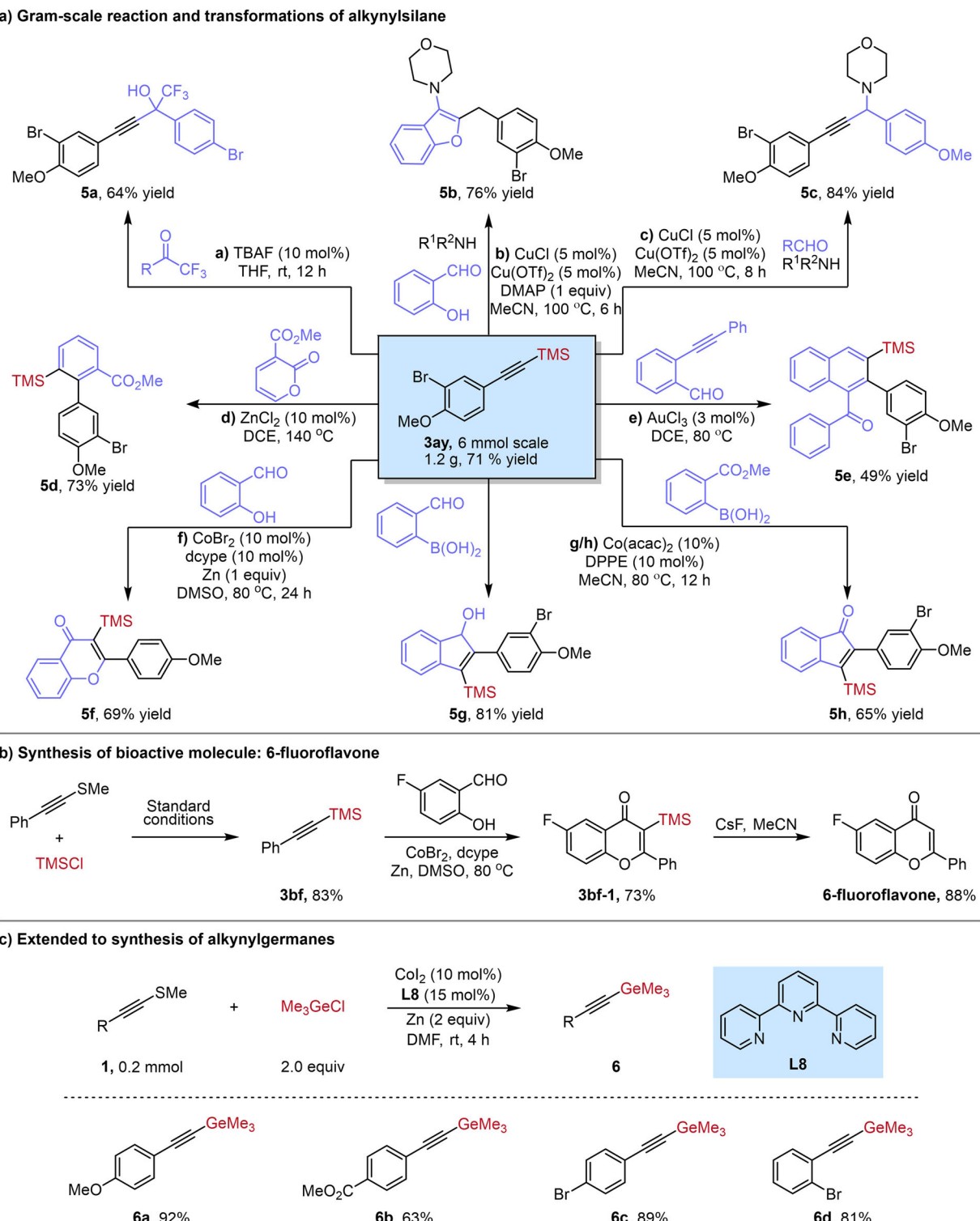

**Fig. 5 | Synthetic applications. a** Gram-scale reaction and transformations of alkynylsilane; **b** Synthesis of 6-fluoroflavone; **c** Extended to synthesis of alkynylgermanes. See Supplementary Information for detailed experimental conditions.

in 89% yield in the absence of chlorosilane, which indicated that oxidative addition of alkynyl sulfide with low valent Co to form Csp-Co species was possible. Moreover, in the absence of alkynyl sulfide, chlorosilane **2 l** failed to dimerize. In addition, adding 3 equivalents of water to the reaction system under standard conditions, the corresponding terminal alkyne **1a-2** was generated in 72% yield. These results further indicated that the low valant cobalt catalyst might activate alkynyl sulfide firstly and generate the Csp-Co species (Fig. 6c). Considering whether our reaction involves alkynylzinc species, we monitored the reaction progress and found that the conversion rate of **1a** was basically consistent with the yield of **3a**, and no corresponding iodide product **1a-3** formation after $I_2$ quenched the aliquots at different time periods (Fig. 6d). This suggests that the alkynylzinc intermediate should not be involved in our reaction. To understand which oxidation state of cobalt reacts with **1a** to initiate the reaction, we performed cyclic voltammetry

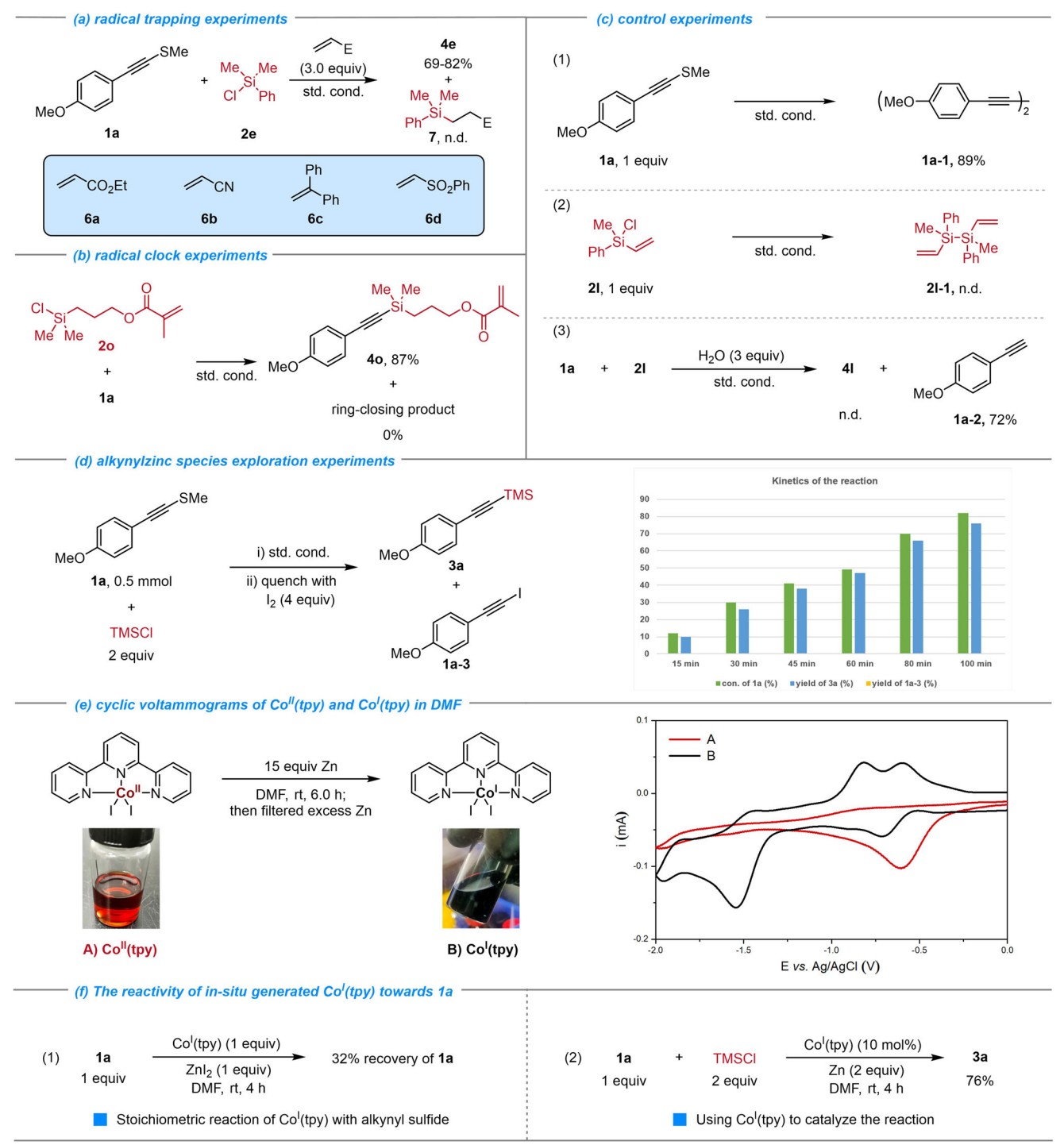

**Fig. 6 | Mechanistic studies. a** Radical trapping experiments. **b** Radical clock experiment. **c** Control experiment. **d** Alkynylzinc species exploration experiment. **e** Cyclic voltammograms of Co^II(tpy) and in-situ generated Co^I(tpy) in DMF. **f** The reactivity of in-situ-formed Co(I) towards alkynyl sulfide **1a**. See Supplementary Information for detailed experimental conditions.

experiments of Co^II(tpy) and in-situ generated low valant Co in DMF. The cyclic voltammogram of **B** revealed a 2-electron reduction wave at E = −1.55 V (Fig. 6e), which corresponds to the Co^I /Co^-1 couple. And that was consistent with Gosmini's observation[16]. Finally, we investigated the reactivity of the in-situ-formed Co^I(tpy) towards alkynyl sulfide **1a**. Both the stoichiometric reaction of Co^I(tpy) with **1a** and using Co^I(tpy) to catalyze the reaction had been carried out (Fig. 6f), and the experimental results indicated that: 1) alkynyl sulfide **1a** can be consumed by equivalent Co^I(tpy); 2) the in-situ-formed Co^I(tpy) is the active intermediate of the reaction.

On the basis of these experimental results and previous reports[24,25,27], a catalytic cycle was proposed in Fig. 7. A low-valent Co^I species was generated in-situ via reduction of Co^II precursor with zinc[60,61]. Then, the alkynyl sulfide **1** reacted with Co^I via oxidative addition to give Csp-Co^III **Int. A**, which was subsequently reduced by Zn to provide a more nucleophilic alkynyl Co^I **Int. B**[62,63]. The S_N2 oxidative addition of chlorosilanes to **Int. B**[64,65], possibly through a five-coordinate cobalt intermediate **C**[25,66,67], to deliver alkynyl Co^III **D**. Reductive elimination of complex **D** would afford the desired product **3** with generation of Co^I to entered next catalytic cycle.

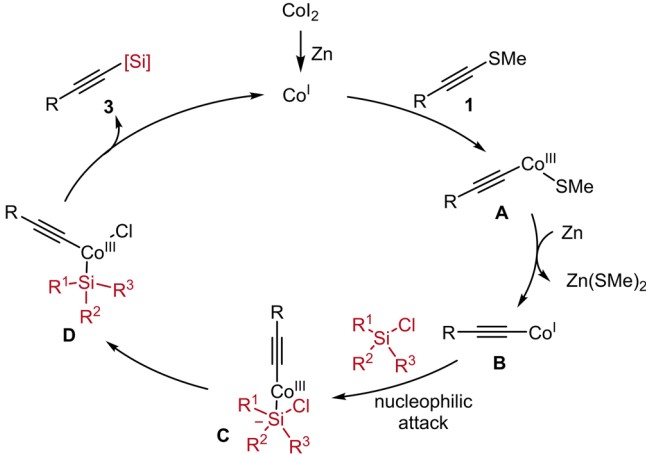

**Fig. 7 | Catalytic cycle.** A plausible mechanism for cobalt-catalyzed cross-electrophile coupling of alkynyl sulfides with unactivated chlorosilanes.

## Discussion

In conclusion, we reported the cobalt-catalyzed cross-electrophile coupling between alkynyl electrophiles with chlorosilanes to construct the C(sp)-Si bonds in the presence of commercially available cobalt catalyst with zinc as the reductant. The utilization of easily prepared alkynyl sulfides by our previous reported work as the alkynyl electrophile was key to achieving cross selectivity with various chlorosilanes. These cobalt-catalyzed XEC between alkynyl sulfides with chlorosilanes exhibited excellent chemoselectivity towards various electrophiles such as, aryl chloride, bromide, iodide, triflate, tosylate and alkyl chloride. Moreover, the efficacy of this approach is further demonstrated through late-stage modification of structurally intricate natural products and drugs. Mechanistic studies revealed that the cobalt catalyst reacts with alkynyl sulfides first and it is likely the chlorosilanes react through a $S_N2$ process via a non-radical pathway. Developing other cross-electrophile coupling of alkynyl sulfides as a stable and practical alkynyl electrophiles are ongoing in our laboratory.

## Methods

### General procedure for cobalt-catalyzed cross-electrophile coupling of alkynyl sulfides with unactivated chlorosilanes

The procedure was conducted in a nitrogen-filled glove box. To a reaction vial equipped with a magnetic stir bar was added $CoI_2$ (6.3 mg, 0.02 mmol), **L8** (7.0 mg, 0.03 mmol), Zn (26.2 mg, 2 equiv). A solution of **1** (0.2 mmol) and chlorosilane **2** (0.4 mmol) in DMF (2.0 mL) was added. The reaction vial was sealed and removed from the glove box. The mixture was stirred at room temperature for 4 h, subsequently quenched with water (10.0 mL) and extracted with ethyl acetate (3 × 15.0 mL). The combined organic layers were washed with water, brine, dried over anhydrous $Na_2SO_4$, and concentrated under reduced pressure. The residue was purified by flash chromatography on silica gel to afford the desired product.

### General procedure for cobalt-catalyzed cross-electrophile coupling of alkynyl sulfides with Me₃GeCl

The procedure was conducted in a nitrogen-filled glove box. To a reaction vial equipped with a magnetic stir bar was added $CoI_2$ (6.3 mg, 0.02 mmol), **L8** (7.0 mg, 0.03 mmol), Zn (26.2 mg, 2 equiv). A solution of **1** (0.2 mmol) and Me₃GeCl (0.4 mmol) in DMF (2.0 mL) was added. The reaction vial was sealed and removed from the glove box. The mixture was stirred at room temperature for 4 h, subsequently quenched with water (10.0 mL) and extracted with ethyl acetate (3 × 15.0 mL). The combined organic layers were washed with water, brine, dried over anhydrous $Na_2SO_4$, and concentrated under reduced

pressure. The residue was purified by flash chromatography on silica gel to afford the desired product.

## Data availability

Data related to materials and methods, optimization of conditions, experimental procedures, mechanistic experiments, and spectra are provided in the Supplementary information. All data are available from the corresponding authors upon request.

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

## Acknowledgements

The authors thank National Natural Science Foundation of China (No. 21971074, L.H.; No. 22001076, L.H.), Natural Science Foundation of Guangdong Province (No. 2019A1515010006, L.H.; No. 2021A1515010159, L.H.) for the financial support. We thank Dr. Matthew J. Goldfogel (Bristol Myers Squibb) for a helpful discussion.

## Author contributions

D.X. and L.H. designed the experiments. D.X. performed experiments. J.L., D.C. and B.H. assisted with melting points determination, some substrates synthesis and validation. D.X. wrote the paper and L.H. revised reviewed & edited the paper. H.J. contributed to discussions. All authors discussed the results and commented on the manuscript. L.H. directed the whole project.

## Competing interests

The authors declare no competing interests.
