## [Peer Review File · Nature Communications]

Cobalt-Catalyzed Cross-Electrophile Coupling of Alkynyl Sulfides with Unactivated ChlorosilanesReviewers' Comments:

Reviewer #1:

Remarks to the Author:

The manuscript by Huang and co-workers describes a new cobalt-catalyzed reductive cross-coupling of alkynyl sulfides with chlorosilanes. Various alkynylsilanes are obtained in good to excellent yields under mild conditions. Many functional groups such as ester, nitrile and amide are present. Although many examples are described, mechanistic studies are too limited. Some further experimental studies will be necessary to gain a better understanding of the mechanism. Why is an organozinc species not formed from an alkynyl sulfide? It has already been shown that an arylzinc species can be formed from a C-SMe with cobalt (Chem Comm 2010). What is the oxidation state of cobalt in the mechanistic cycle?

Overall, considering the standards for publication in Nature communications, this reviewer concludes that the manuscript in its current state is not acceptable for publication in Nature communications.

The following comments should also be addressed:

- 1) Many references do not list all the authors' names, starting with reference 1. In addition, reference 14 is missing a page. Therefore, all the references should be checked.
- 2) Reading this publication, one has the impression that only reductive cross-couplings using nickel have been described. However, I would like to point out that first reductive couplings using non-noble metals were reported by Gosmini using cobalt. References should be added.
- 3) Cobalt has also reacted with thioethers although it was a heteroaromatic.
- 4) Do we not observe cycloaddition reactions with manganese?
- 5) In line 67, the sentence is not clear at all
- 6) Is it necessary to use 15% terpy? What is the yield with only 10% terpy?
- 7) What is the effect of a phosphine as ligand?
- 8) Under these conditions, I am surprised that an ArZnBr was not formed with e.g. the product 3ar!
- 9) In Figure 5, the conditions should be indicated on arrows for better understanding
- 10) The role of ZnI_2 is unclear. Did the authors try to use ZnCl_2 or ZnBr_2 or other salts to avoid complexation of the catalyst by SMe?
- 11) Are they sure that no organozinc species are formed in the medium?
- 12) In SI, product 1b is not pure enough according to the ^{13}C spectrum. The F19 spectrum of 1ax is missing. In the ^1H spectrum of compound 3m, what does the singlet at 4 correspond to? It looks like the same spectrum as 3n

Reviewer #2:

Huang and co-workers describe a cross-electrophile coupling of alkynyl sulfides with chlorosilanes for the synthesis of various alkynylsilanes. The alkynyl-SMe constitutes a suitable alternative to corresponding alkynyl-halides electrophiles in cross-electrophile coupling. Although transition metal-catalyzed cross-electrophile coupling of chlorosilanes has been extensively studied, the substrate range is still limited, especially for unactivated chlorosilanes, which remains an important challenge in organic synthesis. In this study, a broad range of commonly used chlorosilanes are well tolerated. This method exhibited excellent chemoselectivity towards alkynyl sulfides, while various electrophiles, such as aryl chloride, bromide, iodide, triflate, tosylate and alkyl chloride, were unreacted. Overall, the broad substrate scope, mechanistic studies, and proposed catalytic cycle are all validated. Besides, synthetic utility has been demonstrated in the synthesis of various key products, which can significantly simplify the synthesis of complex molecules. This method represents an important advance in the field of cross-electrophile coupling. As such, this reviewer would suggest that this work is potentially suitable for Nature Communications before addressing the following issues by the authors.

1. The role of the zinc salts additive should be further validated. The combination use of Mn and ZnI₂ gave a higher yield (67%) than that of Zn and ZnI₂ (31-33%). What happens if the catalytic loading of ZnI₂ or ZnCl₂ was used in the reaction rather than 1.0 equiv when Mn or Zn is used as a reductant, respectively?
2. This reviewer would suggest adding a column that indicates the yield of 1,3-dimer byproduct in each entry in Table 1. What reaction parameter is the most important for inhibiting this byproduct?
3. Can TIPS-alkynyl sulfide (similar to 1-a) be employed as a substrate in this cross-coupling reaction?
4. In addition to SMe, how about larger groups than the Me group in the reaction?
5. Most of the alkynyl sulfides were prepared from related 1,2,3-thiadiazole; thus, the author should indicate the starting material and their synthetic procedure should be included in the supporting information.
6. For some new products (such as 5a, 5d, 53), corresponding spectra and mass

spectrometry should be provided as much as possible. If the product has been reported in the literature, corresponding references should be indicated.

7. Some typographical and grammatical issues that should be corrected.

1. Line 24 in the manuscript, "is"---"has been"
2. Lines 65-67 and 74-76, are duplicated.
3. Line 78, the word "but" should be deleted.
4. Line 135, "synthetic"---"synthetically"
5. Line 153, "were"---"was"
6. Line 158, "dected"---"detected"
7. Line 174, "commerically"---"commercially"
8. Line 176, "elecrophile"---"electrophile"
9. Line 178, "futhermore" ---"furthermore"

Reviewer #3:

Remarks to the Author:

Cross-electrophile coupling (XEC) is a powerful method to construct molecules. In the manuscript, Huang and co-workers reported cobalt-catalyzed cross-electrophile alkynylation using unactivated chlorosilanes with alkynyl sulfides, providing a wide range of various alkynylsilane products in moderate to good yields. The reaction showed very wide substrate scope and 72 examples were tested. The synthetic potential of the methodology was demonstrated by its application in the modification of drugs. The reaction could be carried out on gram-scale and the products can undergo diverse transformations. In addition, the manuscript and supplementary information were well-prepared and the conclusions in the manuscript were supported by the data in the supplementary information. In my opinion, the manuscript was suitable for publication in *Nat. Commun.* on the condition that the following issues were well addressed.

(1) On page 2, Fig 1a, preliminary exploration, How about the results using bromoalkyne and TMSCl under standard conditions (Cobalt catalyst).

(2) What are the results of the C-Ge alkynylation using alkynyl sulfides and chlorogermanes under standard conditions.

(3) The authors conducted a series of further transformations of alkynylsilane products and modification of drugs. The manuscript should show the synthetic utility of their method in the synthesis of bioactive molecules.

Reviewer #1 (Remarks to the Author):

The manuscript by Huang and co-workers describes a new cobalt-catalyzed reductive cross-coupling of alkynyl sulfides with chlorosilanes. Various alkynylsilanes are obtained in good to excellent yields under mild conditions. Many functional groups such as ester, nitrile and amide are present. Although many examples are described, mechanistic studies are too limited. Some further experimental studies will be necessary to gain a better understanding of the mechanism. Why is an organozinc species not formed from an alkynyl sulfide? It has already been shown that an arylzinc species can be formed from a C-SMe with cobalt (Chem Comm 2010). What is the oxidation state of cobalt in the mechanistic cycle? Overall, considering the standards for publication in Nature communications, this reviewer concludes that the manuscript in its current state is not acceptable for publication in Nature communications. The following comments should also be addressed:

Response: We thank this reviewer for the constructive suggestions to improve the quality of this manuscript. We appreciate it very much. First of all, we double checked whether an organozinc species was formed from an alkynyl sulfide using the quenching experiments as Gosmini's method. In Chem. Commun 2010, the 2-C-SMe bond of benzo[d]thiazole was activated by CoBr₂ with Zn to form corresponding organozinc reagent. We repeated Gosmini's work and did observe the corresponding iodide product by GC-MS. However, for our reaction, we did not observe the corresponding alkynyl iodide product. The special reactivity of 2-C-S bond of benzo[d]thiazole was also observed by J. Cornella (JACS 2019, 141, 1918) for nickel catalyzed cross-electrophile coupling. We cited those two reference for C-SMe bond involved cross coupling in reference 45 and 48 in the revised manuscript.

a) Repeat Corinne Gosmini's work: *Chem. Commun.*, 2010, 46, 5972-5974

b) Quenching experiment using I₂ in our reaction system

To verify the oxidation state of cobalt catalyst for those transformation, we performed the stoichiometric amount of reduction of $\text{Co}^{\text{II}}(\text{tpy})$ with Zn. An obvious color change was observed, which was consistent with the reference that the corresponding Co^{I} was formed. $\text{Co}^{\text{I}}(\text{tpy})$ in DMF (black solution) generated *in-situ* was very unstable and easily reoxidized by air to give $\text{Co}^{\text{II}}(\text{tpy})$ (dark red solution).

We also performed the cyclic voltammety experiments of $\text{Co}^{\text{II}}(\text{tpy})$ and *in-situ* generated low valant $\text{Co}^{\text{I}}(\text{tpy})$ in DMF. The cyclic voltammogram of B revealed a 2-electron reduction wave at $E = -1.55$ V, which corresponds to the $\text{Co}^{\text{I}}/\text{Co}^{\text{I}}$ couple. That was consistent with Gosmini's observation (ACS Catal. 2020, 10, 12819)

($\text{Co}^{\text{II}}(\text{tpy})$ and *in-situ* generated low valant $\text{Co}^{\text{I}}(\text{tpy})$ in DMF)

(Cyclic voltammograms of A and B in DMF)

Finally, we investigated the reactivity of the *in-situ*-formed $\text{Co}^{\text{I}}(\text{tpy})$ towards alkynyl sulfide **1a**. Both the stoichiometric reaction of $\text{Co}^{\text{I}}(\text{tpy})$ with **1a** and using $\text{Co}^{\text{I}}(\text{tpy})$ to catalyze the reaction had been carried out, and the experimental results indicated that: 1) alkynyl sulfide **1a** could be

consumed by equivalent $\text{Co}^{\text{I}}(\text{tpy})$; 2) the *in-situ-formed* $\text{Co}^{\text{I}}(\text{tpy})$ was the active intermediate of the reaction.

On the basis of these experimental results and previous reports, we have revised the possible catalytic cycle in the revised manuscript (Fig. 7). And those control experiments were added into the revised manuscript, and were highlighted in yellow. We hope this explanation and results are satisfied.

1) Many references do not list all the authors' names, starting with reference 1. In addition, reference 14 is missing a page. Therefore, all the references should be checked.

Response: We thank the reviewer for their attention to details and apologize for the oversight. We have carefully checked the all references and these have been fixed in the revised manuscript.

2) Reading this publication, one has the impression that only reductive cross-couplings using nickel have been described. However, I would like to point out that first reductive couplings using non-noble metals were reported by Gosmini using cobalt. References should be added.

Response: We thank the reviewer for pointing out the omission of the original work about reductive

cross-couplings using cobalt. We redescribed it in the introduction of revised manuscript as follows: “Since the pioneering work of Weix and Gosmini using nickel and cobalt catalysis, there have been great achievements in the development of C-C bond-forming reactions using organic halides”. And we have added the following references (*Org. Lett.* **2003**, *5*, 1043-1045; *Angew. Chem., Int. Ed.* **2008**, *47*, 2089-2092; *Angew. Chem., Int. Ed.* **2011**, *50*, 10402-10405; *ACS Catal.* **2020**, *10*, 12819-12827) to 13-16 of the references in the revised manuscript.

3) Cobalt has also reacted with thioethers although it was a heteroaromatic.

Response: We thank the reviewer for pointing this out. The activation of C(sp²)-SMe bond in heteroaromatic thioether with cobalt catalyst for cross electrophile coupling (*Chem. Commun.*, **2010**, *46*, 5972-5974) has been cited in revised manuscript reference 48

4) Do we not observe cycloaddition reactions with manganese?

Response: We thank the reviewer for pointing this out. We don't observe any cycloaddition products with manganese in our reaction.

5) In line 67, the sentence is not clear at all

Response: We thank the reviewer for pointing out this problem. We apologize for the sentence repetition error here in the original manuscript. We have corrected this mistake in the revised manuscript.

6) Is it necessary to use 15% terpy? What is the yield with only 10% terpy?

Response: We thank the reviewer for pointing this out. The use of 15% terpy is not necessary, and there is a slight decrease in yield (80%) when 10% terpy is used.

7) What is the effect of a phosphine as ligand?

Response: We thank the reviewer for pointing this out. We tried some phosphine ligands in this reaction, and found that these phosphine ligands didn't work in this reaction. We added these results to the revised supporting information (Table S3).

Effect of Phosphine Ligand			
			trace	trace	n.d.	trace

8) Under these conditions, I am surprised that an ArZnBr was not formed with e.g. the product 3ar!

Response: We thank the reviewer for pointing this out. First, we think the activity of C(sp)-S is

higher than the aryl-halides in our reaction, which explained the compatibility of those transformation towards a range of electrophiles such as, aryl chloride, bromide, iodide, triflate, tosylate and alkyl chloride. Then, we also investigated the possibility of producing ArZnBr from Br-containing substrate under our reaction system, and the results are shown below:

Not only **1ar**, but also **1ar-1** failed to obtain corresponding iodide products in the CoI_2 with tpy ligand conditions. We speculated that the combination of cobalt and tridentate ligand was not favor for the formation of aryl zinc species from ArBr. The conditions available for organozinc species in the previous literature, either using **ligand-free** reaction conditions, such as a series of elegant work by Gosmini (*J. Am. Chem. Soc.* **2003**, *125*, 3867–3870; *Tetrahedron Letters* **2003**, *44*, 6417–6420; *J. Org. Chem.* **2009**, *74*, 3221–3224; *Chem. Commun.*, **2010**, *46*, 5972–5974; *Chem. Commun.*, **2012**, *48*, 11561–11563...). or using a reaction system of **cobalt and bidentate ligands** (*J. Org. Chem.* **2011**, *76*, 1972–1978; *Inorg. Chem.* **2023**, *62*, 5906–5919...). Therefore, we thought that there is no formation of ArZnBr species in our reaction system.

9) In Figure 5, the conditions should be indicated on arrows for better understanding

Thanks for your advice. We have added these conditions on arrows in the revised manuscript.

10) The role of ZnI_2 is unclear. Did the authors try to use ZnCl_2 or ZnBr_2 or other salts to avoid complexation of the catalyst by SMe?

Response: We thank the reviewer for pointing this out. when Mn was used as a reductant, we tried the following zinc salts and the results are shown below:

additive (1 equiv)	ZnI ₂	ZnBr ₂	ZnCl ₂	Zn(OTf) ₂
yield of 3a	67%	13%	6%	35%

Based on the above experiments, we think that a suitable zinc salt, such as ZnI₂, is very important to bind the equivalent SMe *in-situ* generated in the system to avoid complexation of the catalyst by SMe. And we found an excellent overview of the role of the Zn in a Liebeskind–Srogl reaction (Pure Appl. Chem., 2002, 74, 115) and we added it in reference 44 for the revised manuscript.

11) Are they sure that no an organozinc species are formed in the medium?

Response: We thank the reviewer for pointing this out. We tried to quench the possible intermediate of organozinc species in the reaction system at different time periods, but no corresponding products were obtained. We think that the organozinc species are not formed in the medium. The results are shown below:

At 15, 30, 45, 60, 80, and 100 minutes, an aliquot of the catalytic reaction (0.025 mmol, 250 μ L) was removed and added to a 1-dram vial containing iodine solution (0.1 mmol, 4 equiv) and dodecane (0.025 mmol) as the internal standard. The resulting mixture was diluted with diethyl ether (0.50 mL), mixed, and filtered through a 2-cm silica plug in a Pasteur pipette directly into a GC vial. Samples were analyzed by GC. **1a-3 was not detected.**

Kinetics of the reaction at 15, 30, 45, 60, 80, and 100 minutes after quenching with I₂

We investigated the reaction process and found that the conversion rate of **1a** was basically consistent with the yield of **3a**, and no corresponding iodide product formation after iodine quenched the reaction at different time periods. We added these results in Fig. 6d of the revised manuscript.

12) In SI, product **1b** is not pure enough according to the ¹³C spectrum. The F19 spectrum of **1ax** is missing. In the ¹H spectrum of compound **3m**, what does the singlet at 4 correspond to? It looks like the same spectrum as **3n**

Response: We thank the reviewer for their attention to details and apologise for the oversight. These have been fixed in the revised Supporting Information.

Reviewer #2 (Remarks to the Author):

Huang and co-workers describe a cross-electrophile coupling of alkynyl sulfides with chlorosilanes for the synthesis of various alkynylsilanes. The alkynyl-SMe constitutes a suitable alternative to corresponding alkynyl-halides electrophiles in cross-electrophile coupling. Although transition metal-catalyzed cross-electrophile coupling of chlorosilanes has been extensively studied, the substrate range is still limited, especially for unactivated chlorosilanes, which remains an important challenge in organic synthesis. In this study, a broad range of commonly used chlorosilanes are well tolerated. This method exhibited excellent chemoselectivity towards alkynyl sulfides, while various electrophiles, such as aryl chloride, bromide, iodide, triflate, tosylate and alkyl chloride, were unacted. Overall, the broad substrate scope, mechanistic studies, and proposed catalytic cycle are all validated. Besides, synthetic utility has been demonstrated in the synthesis of various key

products, which can significantly

simplify the synthesis of complex molecules. This method represents an important advance in the field of cross-electrophile coupling. As such, this reviewer would suggest that this work is potentially suitable for Nature Communications before addressing the following issues by the authors.

Response: Thank you for your high recognition of our work.

1. The role of the zinc salts additive should be further validated. The combination use of Mn and ZnI₂ gave a higher yield (67%) than that of Zn and ZnI₂ (31-33%). What happens if the catalytic loading of ZnI₂ or ZnCl₂ was used in the reaction rather than 1.0 equiv when Mn or Zn is used as a reductant, respectively?

Response: We thank the reviewer for pointing this out. When the combination use of Mn and ZnI₂ gave a higher yield (67%), we used tpy as the ligand (manuscript, Table 1, entry 7); and the combination use of Zn and ZnI₂ gave a lower yield (31-33%), we used 4,4'-dmbpy as the ligand in the conditional optimization process (supporting information, Table S1, entry 8 and 9). According to your suggestion, we tried to use the catalytic loading of ZnI₂ or ZnCl₂ in the reaction rather than 1.0 equiv when Mn or Zn is used as a reductant, respectively. The results are shown below:

additive (10 mol%)	ZnI ₂	ZnI ₂	ZnCl ₂	ZnCl ₂
[red] (2 equiv)	Mn	Zn	Mn	Zn
yield of 3a	11%	86%	trace	87%

Based on the above experiments, we think that the equivalent zinc is very important to bind the equivalent SMe *in-situ* generated in the system to avoid complexation of the catalyst by SMe. And we found an excellent overview of the role of the Zn in a Liebeskind–Srogl reaction (*Pure Appl. Chem.*, **2002**, *74*, 115) and we added it in reference 44 of the revised manuscript.

2. This reviewer would suggest adding a column that indicates the yield of 1,3-dimer byproduct in each entry in Table 1. What reaction parameter is the most important for inhibiting this byproduct?

Response: Thanks for your advice. We have added a column that indicates the yield of 1,3-dimer byproduct in each entry in Table 1. As you can see, solvents and catalysts are the most critical factors in inhibiting the generation of byproduct.

3. Can TIPS-alkynyl sulfide (similar to 1-a) be employed as a substrate in this cross-coupling reaction?

Response: Thanks for your advice. We conducted the related experiment and found that TIPS-alkynyl sulfide can't be employed as a substrate in this reaction. And under standard conditions, almost all raw materials remained.

4. In addition to SMe, how about larger groups than the Me group in the reaction?

Response: We thank the reviewer for pointing this out. We synthesized the following substrates for reaction exploration, and the results are shown below. Those results were added into Table S6 in the revised Supporting Information.

Entry	SR	Yield (%) of 3a
1	SMe	87
2	SPh	81
3	$\text{S-CH}_2\text{CH}_2\text{CH}_2\text{CH}_3$	80
4	$\text{S-CH}_2\text{CH}_2\text{CH}_3$	41

5. Most of the alkynyl sulfides were prepared from related 1,2,3-thiadiazole; thus, the author should indicate the starting material and their synthetic procedure should be included in the supporting information.

Response: Thanks for your advice. We have added the starting material and their synthetic procedure in the revised Supporting Information.

6. For some new products (such as 5a, 5d, 53), corresponding spectra and mass spectrometry should be provided as much as possible. If the product has been reported in the literature, corresponding references should be indicated.

Response: We thank the reviewer for pointing these out. We have added corresponding spectra and mass spectrometry in the revised Supporting Information.

7. Some typographical and grammatical issues that should be corrected.

1. Line 24 in the manuscript, “is”---“has been”

2. Lines 65-67 and 74-76, are duplicated.

3. Line 78, the word “but” should be deleted.

4. Line 135, “synthetic”---“synthetically”

5. Line 153, “were”---“was”

6. Line 158, “dected”---“detected”

7. Line 174, “commerically”---“commercially”

8. Line 176, “elecrophile”---“electrophile”

9. Line 178, “furthermore” ---“furthermore”

Response: We thank the reviewer for their attention to details and apologise for the oversight. We have corrected these mistakes in the revised manuscript.

Reviewer #3 (Remarks to the Author):

Cross-electrophile coupling (XEC) is powerful method to construct molecules. In the manuscript, Huang and co-workers reported cobalt-catalyzed cross-electrophile alkynylation using unactivated chlorosilanes with alkynyl sulfides, providing a wide range of various alkynylsilanes products in

moderate to good yields. The reaction showed very wide substrate scope and 72 examples was tested. The synthetic potential of the methodology was demonstrated by its application in the modification of drugs. The reaction could be carried out on gram-scale and the products can undergo diverse transformations. In addition, the manuscript and supplementary information was well-prepared and the conclusions in the manuscript were supported by the data in the supplementary information. In my opinion, the manuscript was suitable for publication in Nat. Commun. on the condition that the following issues were well addressed.

Response: Thank you for your high recognition of our work.

(1) On page 2, Fig 1a, preliminary exploration, How about the results using bromoalkyne and TMSCl under standard conditions (Cobalt catalyst).

Response: We thank the reviewer for pointing this out. Under standard conditions (Cobalt catalyst), we didn't get the corresponding product by using bromoalkyne and TMSCl, and almost the dimer product was obtained because bromoalkyne was too active to undergo homocoupling instead of the cross-coupling.

(2) What's the results of the C-Ge alkynylation using alkynyl sulfides and chlorogermanes under standard conditions.

Response: We thank the reviewer for pointing this out. The C-Ge alkynylation using alkynyl sulfides and chlorogermanes under standard conditions (Cobalt catalyst) proceeded smoothly to give the target products in good yields, and we added these results in Fig. 5c of the revised manuscript.

(3) The authors conducted a series of further transformation of alkynylsilane products and modification of drugs. The manuscript should show the synthetic utility of their method in the synthesis of bioactive molecules.

Response: Thanks for your advice. To further demonstrate the synthetic utility of this newly developed methodology, the efficient synthesis of 6-fluoroflavone, an anti-rhinovirus agent, was carried out using current method as the key step. And we added this synthetic utility in Fig. 5b of the revised manuscript.

Reviewers' Comments:

Reviewer #1:

Remarks to the Author:

I had the opportunity of evaluating this article in a previous submission. Different points I mentioned have been addressed and the revised manuscript has been improved and is now best suited for Nature Communications.

Reviewer #2:

Remarks to the Author:

The authors satisfactorily addressed all the comments. The manuscript is suitable for publication in its current form.

Reviewer #3:

Remarks to the Author:

The authors have addressed all the concerns this reviewer had to the initial submission. It is now ready for publication in Nat Commun.

Point-by-point response to reviewers

Reviewer #1 (Remarks to the Author):

I had the opportunity of evaluating this article in a previous submission. Different points I mentioned have been addressed and the revised manuscript has been improved and is now best suited for Nature Communications.

Response: We highly appreciate this reviewer for taking the time and effort to review our manuscript, and their excellent comments and insightful suggestions, helped to significantly improve the quality of our manuscript.

Reviewer #2 (Remarks to the Author):

The authors satisfactorily addressed all the comments. The manuscript is suitable for publication in its current form.

Response: We thank the reviewer for constructive comments and insightful advice that helped us improve the manuscript.

Reviewer #3 (Remarks to the Author):

The authors have addressed all the concerns this reviewer had to the initial submission. It is now ready for publication in Nat Commun.

Response: We thank the reviewer for the excellent comments and insightful suggestions that helped us improve the manuscript.